# Role of Circadian Transcription Factor Rev-Erb in Metabolism and Tissue Fibrosis

**DOI:** 10.3390/ijms232112954

**Published:** 2022-10-26

**Authors:** Ghulam Shere Raza, Nalini Sodum, Yagmur Kaya, Karl-Heinz Herzig

**Affiliations:** 1Research Unit of Biomedicine, Medical Research Center, Faculty of Medicine, University of Oulu, 90220 Oulu, Finland; 2Department of Nutrition and Dietetics, Faculty of Health Sciences, Marmara University, 34854 Istanbul, Turkey; 3Oulu University Hospital, University of Oulu, 90220 Oulu, Finland; 4Pediatric Gastroenterology and Metabolic Diseases, Pediatric Institute, Poznan University of Medical Sciences, 60-572 Poznań, Poland

**Keywords:** circadian rhythm, metabolism, transcription factor, Rev-Erb, fibrosis

## Abstract

Circadian rhythms significantly affect metabolism, and their disruption leads to cardiometabolic diseases and fibrosis. The clock repressor Rev-Erb is mainly expressed in the liver, heart, lung, adipose tissue, skeletal muscles, and brain, recognized as a master regulator of metabolism, mitochondrial biogenesis, inflammatory response, and fibrosis. Fibrosis is the response of the body to injuries and chronic inflammation with the accumulation of extracellular matrix in tissues. Activation of myofibroblasts is a key factor in the development of organ fibrosis, initiated by hormones, growth factors, inflammatory cytokines, and mechanical stress. This review summarizes the importance of Rev-Erb in ECM remodeling and tissue fibrosis. In the heart, Rev-Erb activation has been shown to alleviate hypertrophy and increase exercise capacity. In the lung, Rev-Erb agonist reduced pulmonary fibrosis by suppressing fibroblast differentiation. In the liver, Rev-Erb inhibited inflammation and fibrosis by diminishing NF-κB activity. In adipose tissue, Rev- Erb agonists reduced fat mass. In summary, the results of multiple studies in preclinical models demonstrate that Rev-Erb is an attractive target for positively influencing dysregulated metabolism, inflammation, and fibrosis, but more specific tools and studies would be needed to increase the information base for the therapeutic potential of these substances interfering with the molecular clock.

## 1. Introduction

Fibrosis is the body’s response to injuries, and chronic inflammation is increasingly recognized as an important cause of morbidity and mortality [1]. It has been estimated that fibrosis in different organs is associated with 45% of all deaths in the industrialized world [2]. Fibrosis is defined as “an excessive accumulation of extracellular matrix (ECM) proteins in tissues, preventing functions and dynamically remodeling” [3,4]. After an injury, molecules from damaged cells induce an inflammatory reaction with the attraction of cells e.g., monocytes, neutrophils, and other immune cells and increased fibroblasts differentiation into ECM-producing myofibroblasts [5]. ECM is a noncellular component essential for tissue differentiation, morphogenesis, and homeostasis and includes fibrous proteins collagens, fibronectins, elastin, laminins, and hydrophilic proteoglycans [6]. During fibrosis, the functional cells such as cardiomyocytes, hepatocytes, acinar cells, podocytes, alveolar cells, and neurons are being replaced by ECM proteins such as collagen and fibronectin in the tissues [3,7]. Various factors such as hereditary disorders, age, obesity, diabetes and hypertension, hypercholesterolemia, persistent infections, and recurrent exposure to toxins, irritants/smoke affect wound healing after injury, resulting either in progressive fibrosis or healthy tissue repair [8]. Collagen is the most abundant fibrous protein of ECM, which provides tensile strength, regulates cell adhesions, and supports chemotaxis and tissue development [9]. Myofibroblasts synthesize ECM proteins and are well-known fibrosis effectors in various tissues (Figure 1) [10].

Myofibroblasts are spindle or stellate shape fusiform cells with a similar ultrastructure as smooth muscle cells and are typically found in scar and granulation tissues and stroma of tumors [11]. These cells possess both secretory characteristics of fibroblasts and contractile characteristics of smooth muscles and play vital roles in swiftly repairing injured tissues. Myofibroblasts lack smooth muscle markers desmin and myosin and display elevated levels of stress fibers, exosomes, and microvesicles [12]. They typically contain elongated and serrated nuclei, multiple dendritic processes, extensive rough ER, fibrillar collagens, and microfilament proteins [13,14]. Myofibroblasts synthesize and secrete ECM proteins such as collagens (e.g., I and III), a specialized isoform of fibronectin, periostin, and cadherin-2, and -11 [13]. In addition, they secrete numbers of cytokines, inflammatory mediators (CO, H_2_O_2_, NO, O-radical, HETE, PAF, and prostacyclin), and growth factors (IGF, PDGF TGF-β) maintaining the inflammatory response to an injury [15]. Cell-adhesion molecules (CAMs) such as ICAM1 and VCAM1 are expressed in subtypes of myofibroblasts, and induce communications with immune cells such as T-cells and neutrophils [16,17]. 

Transdifferentiation of quiescent mesenchymal cells to activated myofibroblasts is a key event in the pathogenesis of fibrosis [18]. The potential progenitor cells for myofibroblasts consist of fibroblasts, fibrocytes, epithelial and endothelial cells, and other mesenchymal cells such as pericytes, depending on the insult and organ [19,20]. The early changes occur during myofibroblast formation with mitochondrial changes and ROS production [21,22]. Recently, it has been demonstrated that mitochondrial Ca^2+^ signaling is a key regulator of myofibroblast differentiation and fibrosis [21]. The differentiation of myofibroblasts is initiated by various factors such as hormones (endothelin-1 and angiotensin II), growth factors (TGF-β and VEGF), and inflammatory cytokines (TNF-α, IL-1, and IL-6), Ca^2+^, and mechanical stress [23,24], which are linked to the circadian rhythm [25]. 

## 2. Circadian Rhythm

Circadian rhythm is an autonomous self-sustained oscillation of 24 h, which regulates the metabolism of organisms [26,27]. The circadian clock drives the daily rhythm and controls coordinated oscillations of many neuroendocrine, signaling, and metabolic pathways [28]. Disturbances of these interactions, as in shift work, transcontinental fights, and irregular eating patterns, are increasingly recognized as risk factors for CNS, metabolic, cardiovascular diseases, and cancer (Figure 2) [29,30,31].

At the cellular level, circadian clock genes consist of transcription-translation feedback loops, in which *Clock* and brain and muscle Arnt-like protein-1 (*Bmal1*) drive expression of 2 negative-feedback arms controlled by Period/Cryptochrome (*Per*/*Cry*) and the two paralogs, *Rev-Erbα* and *Rev-Erbβ* [32]. In turn, these negative-feedback arms repress Bmal1/Clock transactivation function (Per/Cry) or Bmal1 expression (Rev-Erbα/β) and Bmal1 activation (ROR) (Figure 3). 

Circadian rhythms are influenced by light, physical activity, diet, aging, and disease states [33,34]. Genetic polymorphisms within the core clock genes *Bmal1* and *Clock* are associated with metabolic diseases such as obesity, type 2 diabetes, and hypertension [35,36]. More than half of human genes display circadian oscillations in at least one body tissue or organ [37]. Changes in circadian transcriptions affect major homeostatic mechanisms, including stem cell regulation, mitochondrial function, and immune responses. Mitochondria exhibit a bidirectional relationship with the circadian clock via a reciprocal interaction between peroxisome proliferator-activated receptor gamma co-activator 1-alpha (PGC-1α) and *Bmal1* [38]. In addition, fatty acid oxidation enzymes and electron transfer flavoproteins also follow circadian oscillation [39]. Furthermore, clock-controlled genes (CCGs) such as O-linked beta-D-N-acetylglucosamine (*O-GlcNAc*) signaling, nicotinamide adenine dinucleotide (NAD+)-dependent sensors, nicotinamide phosphoribosyl transferase (NAMPT), silent mating type information regulation 2 homolog 1 (*SIRT1*), and 5′-adenosine monophosphate-activated protein kinase (AMPK) link the clocks with metabolic regulators [40,41,42]. Nuclear receptors (NRs) exhibit circadian oscillations in metabolic tissues, in addition to direct transcriptional regulation of *Bmal1* via *ROR* and *Rev-Erb.* Circadian clocks regulate various immune markers such as interleukin IL-2, IL-10, IL-6, IL-1β, and tumor necrosis factor-α (TNF-α) [43,44]. Disruption of the circadian rhythm increases inflammatory markers such as IL-6 and TNF-α, affecting circulating lymphocytes and natural killer (NK) cells [45]. Inflammation-related genes such as nuclear factor erythroid2-related factor 2 (*NRF2*), nuclear factor kappa B (NF-*κ*B), chemokine ligands, and PPARs are strongly linked with the molecular clocks [46,47,48].

### Circadian Repressor Rev-Erb

*Rev-Erbα* (*Nr1d1*) and *Rev-Erbβ* (*Nr1d2*) are members of the nuclear receptor (NRs) superfamily and have significant roles in the maintenance of circadian rhythm, metabolic process, and immune functions [49,50]. They have distinct features among NRs because their ligand binding domain lack activation function-2 (AF2) at C-terminus, which is essential for interacting with co-activators and gene transcription [51]. *Rev-Erb* acts as constitutive repressors of circadian transcription either passively by competing with the transcriptional activator retinoic acid-related orphan receptor (ROR) at binding elements (RORE) [52,53], or actively by recruiting the NCoR-HDAC3 corepressor complex [54,55]. Rev-Erbs regulate glucose, lipids, and energy metabolism as well as adipogenesis and inflammation (Figure 4) [56,57].

Synthetic Rev-Erb agonists modulate metabolic states by altering central and peripheral clocks [49,58]. The Rev-Erb agonist (SR9009) reduced obesity and improved plasma lipid profiles in DIO mice by increasing energy expenditure [58,59]. In addition, the Rev-Erb agonist had beneficial effects on jet lag, sleep disturbances, metabolic diseases, inflammation, and cancer [60,61,62]. Daily administration of the Rev-Erb agonist (SR9011) reduced weight gain, fat mass, and expression of lipogenic genes in normal mice [58,63]. Gibbs et al. showed that genetic knockdown of *Rev-Erbα* increased IL-6 expression in response to lipopolysaccharides (LPS) and the Rev-Erbα ligand (GSK4112) inhibited the production and release of IL-6 in human macrophages [62]. Furthermore, the author reported that in the absence of *Rev-Erbα*, the inhibitory effect of GSK4112 on IL-6 response was abolished, indicating that Rev-Erbα regulates the production and secretions of inflammatory cytokines [62].

## 3. Fibrotic Mediators 

Two major mediators of tissue fibrosis are transforming growth factor-β (TGF-β) [64] and hypoxia-inducible factor 1 alpha (HIF-1α) [65]. Proteases such as plasmin, cathepsin, thrombospondin, calpain, integrin-*α*v*β*6, and MMPs activate TGF-β, which is a potential target for antifibrotic drugs [66,67]. TGF-β affects the circadian clock gene machinery by inducing expression of *Bmal1* and *Clock* and inhibiting expressions of *Per1/2*, *Rev-Erbα*, *RORα*, and cold-inducible RNA binding protein (*Cirbp*) (Figure 5) [68,69]. TGF-β2 inhibits the expression of circadian clock genes; *Per*, *Rev-Erbα,* and CCGs; D-site albumin promoter binding protein (DBP) and thyrotroph embryonic factor (*TEF),* without altering *Bmal1* expression in NIH3T3 fibroblasts and HT22 neurons [70]. TGF-β signaling is influenced by *Rev-Erb* and *Bmal1*, and TGF-β itself alters *Rev-Erb* and *Bmal1* expressions [71,72]. Hypoxia increases the expression of proteases such as MMPs and thrombospondin 1 (TSP-1) activating TGF-β [73].

HIF-1α activation increases the production of proinflammatory cytokine TNF-α, IL-1β, IL-6, and IL-8, while HIF-2α reduces cytokine productions [74,75]. Clock genes respond to acute hypoxia in a tissue-specific manner with an inter-tissue clock misalignment [76]. HIF-1α activation disrupted the circadian rhythm by dampening the oscillation amplitudes of *Bmal1* and *Per2* in a dose-dependent manner in myoblasts and osteosarcoma (U2OS) cells [39,77]. PER2 enhances HIF-1α activity by facilitating the recruitment of HIF-1α to the hypoxia-response element (HRE) on the VEGF promoter [78]. *Cry1* inhibits HIF-1α transactivation and reduces the half-life of HIF-1α [79]. 

Other important molecular pathways for tissue fibrosis include platelet-derived growth factor (PDGF), connective-tissue growth factor (CTGF), vasoactive peptides (especially angiotensin II and endothelin-1), and integrins [64,80]. TNF-α, CTGF, PDGF, and FGF induce fibrosis by increasing ECM accumulation via myofibroblast differentiation [81]. In addition, various cytokines such as IL-4, IL-13, IL-17, IL-25, and IL-33 induce fibrosis in different organs [82,83,84,85].

The role of Rev-Erb in different organs will be discussed in the following chapters.

## 4. Effect of Rev-Erb in Tissue Fibrosis

### 4.1. Rev-Erb in Heart Fibrosis

The prevalence of heart diseases are continuously rising, and globally, approximately 64.3 million people are affected by heart failure (HF) [86]. Recently, a meta-analysis showed that the prevalence of ‘all types’ heart failure in developed countries is around 11.8% in those aged 65 years and older [87]. Ischemic injury leads to cardiac remodeling and fibrosis, resulting in HF. Cardiac injuries lead to the development of cardiac fibrosis, which is associated with increased ECM proteins in the myocardium. The ECM network of the heart mainly consists of collagen 1 (Col1; 85% of the total myocardial collagen), providing tensile strength with minor amounts of collagen III (Col3), which provides elasticity [88]. The fibrotic response is triggered by growth factors and proteases in the cardiac ECM after injury [89]. 

Rodents and human hearts displayed robust circadian rhythms [90,91]. Studies in murine models demonstrated that genetic disruption of the circadian clock gene *Bmal1* resulted in cardiomyopathy and reduced lifespan [92,93]. Clock-deficient mice developed age-dependent cardiac hypertrophy and interstitial fibrosis at 21 months of age [94]. Furthermore, shift workers are more prone to cardiovascular diseases such as atrial fibrillation and coronary heart disease (CHD) [95]. 

Rev-Erb agonist (SR9009) prevented cardiomyocyte hypertrophy, reduced fibrosis, and halted the progression of HF in mice [96]. In addition, the Rev-Erb agonist (SR9009) improved left ventricles (LV) function and survival after myocardial infarction (MI) (Table 1) [97]. SR9009 decreased the expression of cytokines IL-6, monocyte chemoattractant protein-1 (MCP-1), MMP9, and immune cells (neutrophil and proinflammatory macrophages) infiltration in the infarcted heart [97]. In pressure-induced cardiac hypertrophy by transverse aortic constriction (TAC) in mice the Rev-Erb agonist (SR9009) reduced protein kinase B (AKT) expression and cardiac hypertrophy [94]. In LDL-receptor-deficient mice fed with a western diet, Rev-Erb agonist (SR9009) reduced atherosclerosis [98]. The SR9009-treated mice had a reduced ratio of proinflammatory M1 macrophages to the anti-inflammatory M2 macrophages, indicating that the reduction in atherosclerotic plaque is due to the anti-inflammatory activity of the Rev-Erb agonist [98]. 

Overall, the effects of Rev-Erbs on cardiac functions are only sparsely studied and would need further investigation. In addition, human studies regarding the role of the circadian clock in cardiac fibrosis are lacking.

### 4.2. Rev-Erb in Lung Fibrosis

Inflammatory lung diseases such as COPD and asthma frequently show daily variations in symptoms, and genes involved in lung functions display circadian rhythms [124,125]. Circadian clock genes play an important role in pulmonary fibrosis, COPD, and lung cancer [99,126]. *Bmal1* deletion in lung epithelium enhanced the inflammatory reaction in response to cigarette smoke and lipopolysaccharide (LPS) [102,127]. Dong et al. (2016) reported that TGF-β induction increased *Bmal1* and decreased *Rev-Erbα* and *RORα* expression in the lung while silencing *Bmal1* reduced TGF-β–induced endothelium mesenchymal transition (EMT) and MMP9 production in lung epithelium [128]. Recently, Cunningham et al. demonstrated that circadian repressor Rev-Erbα inhibited myofibroblast differentiation and collagen secretion in cultured fibroblasts and lung tissues of pulmonary fibrosis patients (Table 1) [99]. Mice with emphysema had lower expression of Rev-Erbα genes and proteins in the lung tissues [102]. *Rev-Erb^−/−^* mice displayed increased pulmonary myofibroblast activating markers such as collagen-1 and α-SMA [99]. Sundar et al. demonstrated that Rev-Erbα knockout mice had an increased neutrophils influx in the lungs and proinflammatory cytokines (IL-6, MCP-1, keratinocyte chemoattractant (KC)) release after exposure to cigarette smoke or LPS [100]. Smokers and COPD patients showed lower *Rev-Erbα* mRNA and protein expression compared to nonsmokers in PBMCs, sputum, and lung tissues [101]. The Rev-Erb agonist (GSK 4112) suppressed TGF-β–induced fibroblast differentiation by inhibiting activation of mesenchymal markers in human fetal lung fibroblast 1 cells [115] and attenuated both LPS and cigarette smoking-induced inflammatory responses and pulmonary fibrosis [114,127] in human small airway epithelial cells and mouse lung fibroblasts [100]. 

### 4.3. Rev-Erb in Liver Fibrosis

Liver fibrosis results from a chronic liver injury, e.g., alcohol or overnutrition, which may lead to liver cirrhosis and hepatocellular carcinoma (HCC). The fibrous tissues disrupt the normal architecture and functions of the liver and increase portal vein pressure [129]. Apoptotic hepatocytes activate quiescent hepatic stellate cells (HSCs) to transdifferentiate into myofibroblasts via TGF-β [130,131]. The activated HSCs induce ECM deposition in liver tissues via tissue TIMPs, which upregulate TNFα and IL-1β expression [132]. During liver injury, expression of inflammatory cytokines such as TNF-α, TGF-β, and platelet-derived growth factor (PDGF) increase, which exacerbate the progression of liver fibrosis by activation of the Ras-MAPK, PI3K-AKT/PKB, and PKC pathways [133].

The liver is the main organ for the control of lipid homeostasis, which is disrupted in clock mutant mice [134,135]. In mice liver, app. 17% of lipids display robust circadian rhythmicity [136], with similar results obtained in a human lipidomic study [137]. In the liver, Rev-Erbα regulates multiple genes involved in metabolic pathways [54]. *Rev-Erbα* knockout mice exhibit increased plasma VLDL concentrations and APOC-III expression [103]. Hepatic triglyceride levels were increased in the liver-specific knockout of *Rev-Erbα/β* mice fed with HFD [105]. These mice had an enhanced rhythmic expression of sterol regulatory element binding transcription factor 1 (*Srebf1*), which regulates de novo lipogenesis by its target genes [105]. Similar results on *Srebf1*, cholesterol, and bile acid metabolism were reported in whole-body *Rev-Erbα* knockout mice [138]. *Rev-Erbα* null mice had hepatic steatosis and reduced bile acid synthesis, which were aggravated by the additional knockdown of *Rev-Erbβ* [55,104]. The disrupted circadian behavior was more distinct in the *Rev-Erbα/β* double-knockout model, suggesting additive functions of *Rev-Erbα* and *Rev-Erbβ* [109]. *Rev-Erbα* expression is upregulated in activated HSCs, and an increased hepatic mRNA and protein expression were found in CCL4-treated mice [120,139]. Primary hepatocytes from *Rev-Erbα^−/−^* mice had markedly increased lipid accumulation and *Cyp4a10* and *Cyp4a14* expression [106]. *Cyp4a10* and *Cyp4a1*4 are highly expressed in the liver converting arachidonic acid to 20-hydroxyeicosatetraenoic acid, which induce inflammation via ROS generation [140]. *Cyp4a14* deficient mice had reduced lipid accumulation, hepatic inflammation, and fibrosis in methionine and choline-deficient diet-induced nonalcoholic steatohepatitis (NASH) [113], indicating that *Rev-Erbα* acts as a transcriptional repressor of *Cyp4a10* and *Cyp4a1*4.

In addition to genetically modified animals, synthetic Rev-Erb ligands influenced lipid metabolism. The Rev-Erb agonist (SR9009) displayed beneficial effects such as weight loss and reduced plasma triglycerides and cholesterol [63]. The Rev-Erb agonist (SR9009) inhibited cholesterol biosynthesis by suppressing liver *Hmgcr* and *Srebf2* [116]. In vitro and in vivo studies demonstrated that Rev-Erbα agonist reduced fibrosis in rat HSCs and CCL4-induced liver fibrosis [120,121]. The Rev-Erb agonist (SR9009) inhibited HSCs proliferation by inhibition of the AKT/mTOR/P70S6K pathway [117]. Furthermore, Rev-Erb agonist (SR9009) prevented alcohol-induced liver injury in mice by downregulating *Cyp4a1* expression [106]. Rev-Erb agonist (SR9009) reduced hepatic inflammation (*IL-1α*, *IL-1β*, *Ifnγ*, and *TNFα*) and fibrosis [118] via decreased expression of profibrotic genes (*TGFβ, STAT1 Col3A1*, *Acta2*, *MMP13* and *TIMP1* in a NASH mouse model (*ob/ob* mice fed a high-fat, high-fructose and high cholesterol diet)) [141]. Rev-Erb agonist (SR9009) protected against CCL4-induced liver fibrosis in mice by upregulation of the expression of the clock genes *Bmal1*, *Clock*, *Per2*, *Cry1*, and *RORα* [120]. In NAFLD patients, studies of Rev-Erbα are sparse; however, a significant down-regulation of Rev-Erbα was reported in pediatric patients [142], suggesting that Rev-Erb agonist could be a novel target for the prevention of liver fibrosis. 

### 4.4. Rev-Erb in Adipose Tissue Fibrosis

Adipocyte hypertrophy and hyperplasia are characteristic features of obesity, which trigger inflammation and, eventually, adipose tissue fibrosis [143]. White adipose tissue (WAT) is a complex organ that includes a number of cell types in addition to adipocytes, such as preadipocytes, endothelial cells, fibroblasts, and immune cells (macrophages, mast cells, B and T cells) [144]. An increase in adipocyte size during obesity results in hypoxia due to inadequate blood supply and increased oxygen consumption [145,146]. Hypoxia increases the expression of genes such as vascular endothelial growth factor (*VEGF*) and *MMP2* and *MMP9* in AT affecting vascularization, ECM remodeling, and fibrosis [147]. Enlarged adipocytes are more dysfunctional and produce a higher amount of fatty acid, TNFα, IL-6, IL-8, and MCP-1 and display reduced mitochondrial size [148,149]. Inflammation in AT leads to obesity-associated fibrosis, and macrophages play a major role in WAT fibrosis by Toll-Like Receptor 4 (TLR4) activation [5,150]. TLR4 activation stimulates TGFβ-1 and ECM production as well as fibroblast differentiation by macrophage-inducible C-type lectin [150,151]. The enlarged adipocytes trigger the recruitment of immune cells driving AT inflammation and fibrosis [152,153,154]. 

Several studies demonstrated that functional peripheral clocks are present in various adipose depots of animals and humans [155,156]. Disruption of the circadian rhythm by constant light exposure increased body weight gain, despite similar caloric intakes and total activity in mice [157]. Visceral AT in diet-induced or genetically obese mice had a dampened circadian rhythm with metabolic dysfunctions [158,159]. In addition, obese and diabetic mice (*db/db*) displayed disrupted rhythm and higher expressions of collagen I, III, and VI in AT compared to control mice [160]. Collagen VI (Col6) is a major ECM protein in adipose tissue of humans and mice and accumulates at higher rates under obese and diabetic conditions [161,162,163]. Col6α3 elevates systemic inflammations and causes insulin resistance and fibrosis in adipose tissue by inducing TGFβ-1 expression [162,164]. *Clock* knockout mice displayed visceral adiposity with AT hypertrophy [135,165]. AT-specific deletion of *Bmal1* or *Cry* in mice resulted in higher adiposity, hypertrophic adipocytes, and crown-like structures under HFD [166]. 

*Rev-Erbα* expression was upregulated during adipogenesis in 3T3L1 cells and the synthetic Rev-Erb agonist induced adipocyte differentiation [122]. *Rev-Erbα* knockout mice displayed adiposity and hypertrophied adipocytes, which was aggravated by HFD feeding [107,108]. These mice did not show hyperglycemia due to increased uptake of fatty acid in the muscle by increased lipoprotein lipase (*Lpl*) gene expression in adipose tissue and muscles [108]. *Rev-Erbα* downregulated the triglyceride synthesis enzymes *Lpl* and fibroblast growth factor-21 (FGF21) [108,167]. In *Rev-Erb^−/−^* mice, increased plasma adiponectin and its expression in WAT were found [107]. These results suggest that Rev-Erbs improve insulin sensitivity. Mice with adipocyte-specific knockout of *Rev-Erbα* developed obesity only under a high-fat diet, indicating that Rev-Erbα regulates WAT metabolism in a state-dependent manner [168]. These results suggest that Rev-Erbα does not enforce rhythmic repression of metabolic circuits under basal conditions but rather affects tissue responses to the altered metabolic states. *Rev-Erbα* expression was decreased in both omental adipocytes and adipocytes of obese subjects due to reduced *Bmal1* binding [159]. *Rev-Erbα* expression was increased in subcutaneous fat after weight loss in overweight subjects [169]. The Rev-Erb agonist (SR9011) reduced fat mass and lipogenic gene expression in mice liver [58]. In addition, Solt et al. confirmed that the Rev-Erb agonist (SR9009) reduced adiposity and inflammation in diet-induced obese mice [58]. 

These studies demonstrate that Rev-Erb*α* alleviates metabolic alterations in adipose tissue by reducing inflammation.

### 4.5. Rev-Erb in Skeletal Muscle Fibrosis

Skeletal muscle (SM) accounts for 30–40% of total body mass and regulates whole-body metabolism and energy homeostasis. ECM contains myofibers, nerves, and blood vessels in SM and constitutes up to 10% of the SM weight [170,171]. The major roles of ECM in SM include force transmission, repair, and maintenance of muscle fibers after injury [172]. SM has a high regeneration ability, and SM clocks affect proteostasis, lipid metabolism, and muscle functions [173]. Hodge et al. [174] reported that of 1628 circadian genes in SM, 62% had metabolic roles, mainly in carbohydrate metabolism. Clock disruptions in SM aggravated metabolic dysfunction and muscle atrophy [175,176]. Muscle-specific *Bmal1* knockout mice showed impaired glucose uptake and reduced glucose oxidation in SM without change in glucose tolerance [177], suggesting that reduced muscle insulin sensitivity was most likely compensated by other insulin-sensitive tissues [178]. 

*Rev-Erb* is highly expressed in SM with similar oscillations as in the liver and AT [56,179]. Rev-Erbα regulates muscle mass and muscle fiber type distribution and is selectively expressed in glycolytic type IIB and intermediate type IIA muscle fibers [180]. *Rev-Erbα^−/−^* mice displayed substantially reduced muscle mass with increased expression of atrophy-related genes (atrogenes) [110]. In addition, *Rev-Erbα/β* double-knockout mice exhibited altered circadian wheel-running behavior [109]. *Rev-Erb^−/−^* mice showed reduced running capacity due to reduced mitochondrial biogenesis and functions by deactivating the serine/threonine kinase 11 (STK11)-AMPK-SIRT-1-PGC-1α signaling pathway [59]. The regenerative capacity of muscle was reduced in *Rev-Erb^−/−^* mice; however, partial loss of *Rev-Erbα* increased regenerative capacity in response to injury [111]. *Rev-Erbα* overexpression increased exercise capacity, mitochondrial content, and activity in C2C12 myocytes via AMPK [59] and reduced dexamethasone-induced atrophy-related genes in myoblast cells and loss of muscle mass in mice [110]. *Rev-Erbβ* overexpression affected lipid metabolism and energy expenditure in SM by downregulating *Cd36*, *Fabp-3*, and *-4* mRNA and upregulating *IL-6* expression in mouse myogenic C2C12 cells [181]. *Rev-Erbα* overexpression alleviated autophagy and reduced oxidative stress in skeletal muscles by directly inhibiting genes such as *Ulk1* and perkin (*Park2*) involved in mitochondrial clearance and mitophagy [59,182]. Welch Billion et al. [123] reported that the Rev-Erb (antagonist SR8278) improved mitochondrial biogenesis and muscle functions, and fibrosis in dystrophic mice by activating Wnt signaling. The group suggested differences in mitochondrial biogenesis in fully differentiated versus proliferating muscles using a dystrophic mouse model [123]. 

### 4.6. Rev-Erb in Kidney Fibrosis

Fibrosis is present in chronic kidney disease (CKD) and is characterized by excessive deposition of ECM in the tubular interstitium and glomeruli. CKD is one of the most prominent causes of death and suffering globally [183,184], affecting about 850 million individuals worldwide [185]. Type 2 diabetes and ischemic/hypertensive nephropathy are the two most common causes of CKD in developed nations. Factors that contribute to CKD development include, e.g. xenobiotics, toxins, infections, mechanical obstruction, and autoimmune diseases [186]. The renal medulla is especially prone to hypoxia with increased expression of fibrillar Col I and Col III but also contains Col IV, V, heparan, fibronectin, and laminin [187,188]. The cellular mechanism of renal fibrosis is not yet clear, and there is no specific therapy to halt its progression. In addition, the origin of myofibroblasts in the kidney is controversial and considered to arise from different cell types such as fibroblasts, endothelium, epithelium, and podocytes [189,190]. 

The kidney is under the control of an intrinsic clock, affecting blood flow, glomerular filtration rate (GFR), water, and electrolyte transport [191,192,193]. Partial nephrectomy (5/6Nx) caused glomerulosclerosis, interstitial fibrosis and induced *Bmal1*, *Clock*, and *Rev-Erbα* expression [194]. *Rev-Erbα* and *Rev-Erbβ* were downregulated, and CCGs related to vascular integrity, endothelial function, inflammation, and thrombogenesis were severely deregulated in calcified aortas of rats in CKD [195]. Cisplatin-induced increases in blood urea nitrogen (BUN), serum creatinine, kidney injury molecule (KIM-1), and neutrophil gelatinase-associated lipocalin (NGAL1) were reduced in *Bmal1* knockout mice [196]. Kidney fibrosis induced by ureteral obstruction was ameliorated in *Bmal1* deficient mice [197,198]. *Clock*-deficient mice showed severe kidney fibrosis via activation of *cyclooxygenase 2* in the unilateral ureteral obstruction model of kidney injury [199]. In *Clock* mutant mice, adenine-induced CKD was exacerbated by increased MMPs and 2,8-dehydroxyadenine, and fibrosis markers deposition [200]. In contrast, clock mutant mice (*Clk/Clk* deletion of exon 19 in the Clock locus) ameliorated kidney injury by increased GFR and reduced creatinine, BUN, and fibrosis markers in partially nephrectomized (5/6Nx) mice [201]. The different results on the influence of clocks on kidney functions could be due to various methodologies of acute kidney injury (AKI) and CKD. 

*Rev-Erbα^−/−^* and *Rev-Erbβ^−/−^* mice showed a decreased sensitivity to folic acid-induced AKI and diminished disease severity [112]. The authors reported that Rev-Erbα/β inhibition reduced folic-acid-induced inflammatory cytokine (TNFα, IL-1β) secretion, renal injury (KIM-1 and NGAL), and profibrotic (TGF-β, αMSA) gene expression [112]. *Rev-Erbα* expression was increased in the renal cortex and medulla of spontaneously hypertensive rats (SHRs) and stroke-prone SHR rats (SHRSP), compared to normal Wistar rats [202]. Dexamethasone administration to tubular epithelial cells of mice increased *Rev-Erbα*, Fragile X mental retardation autosomal homolog 1 (*Fxr1*), and phosphoribosyl pyrophosphate amidotransferase (*Ppat*) expression [202,203]. Fxr1 and Ppat are involved in the formation of circadian rhythm in kidney tubules [202]. Rev-Erb antagonist (SR8278) reduced kidney damage in wild-type mice by increasing gene and protein expressions of solute carrier family 7 member 11 (*Slc7a11*) and heme oxygenase (*HO1*), which were elevated in *Rev-Erbβ^−/−^* mice [112]. These studies demonstrate the significantly emerging role of Rev-Erb in the kidney, but it is not yet well understood and needs further investigations.

## 5. Summary and Conclusions

Circadian rhythms are self-controlled oscillations in all organs regulating body functions by directly driving the cyclic expression of nuclear receptors (NRs) and their ligands. As members of this superfamily, the circadian repressor Rev-Erbs regulate energy metabolism, inflammation, and fibrosis. Rev-Erb activation is therapeutically beneficial for alleviating tissue fibrosis in various organs such as the liver, heart, and lungs. The role of Rev-Erb in adipose tissue fibrosis is not fully understood and needs further investigation. Preclinical studies demonstrated the anti-inflammatory and antifibrotic effects of Rev-Erbα; however, limited results on inflammation and tissue fibrosis in clinical settings have been reported. 

Rev-Erbs are promising therapeutic targets for the treatment of cardiometabolic diseases and tissue fibrosis, but this is challenging for drug development due to offsite targets. Several Rev-Erb agonists and antagonists were developed, but currently, there is no specific agonist or antagonist for Rev-Erbα, which is the major player in circadian rhythm regulations compared to Rev-Erbβ. More mechanistic and clinical studies would be necessary to further investigate the roles of Rev-Erbs in fibrotic conditions. Target-specific compounds selective to Rev-Erbα are needed for optimal benefits. Rev-Erb synthetic ligands have poor pharmacokinetic parameters, which hinder the development of these agents. Human studies are required to determine whether modulation of Rev-Erbα/β by synthetic ligands in different organs is beneficial in terms of alleviation of cardiometabolic diseases, inflammation, and tissue fibrosis. 

## Figures and Tables

**Figure 1 ijms-23-12954-f001:**
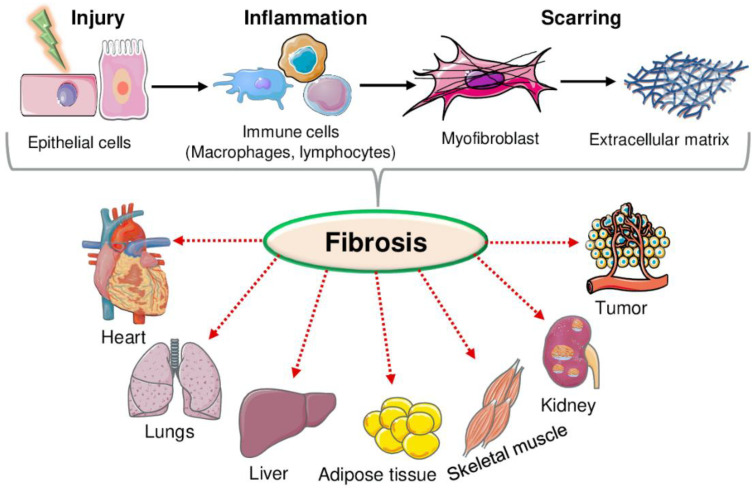
After injury, damaged cells recruit inflammatory cells (e.g., monocytes, neutrophils, and other immune cells), which promote fibroblasts differentiation into ECM-producing myofibroblasts, leading to fibrosis in various tissues.

**Figure 2 ijms-23-12954-f002:**
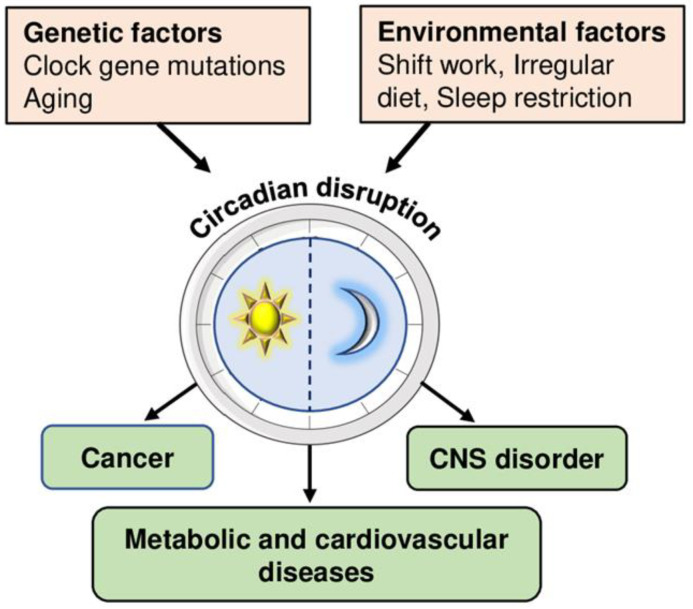
Genetic factors (age and mutations in clock genes) and environmental factors (sleep disturbances, irregular diet, and shift work) disrupt circadian rhythm. Disturbances in the oscillations of the circadian clock lead to cardiometabolic diseases, CNS disorders, and cancer.

**Figure 3 ijms-23-12954-f003:**
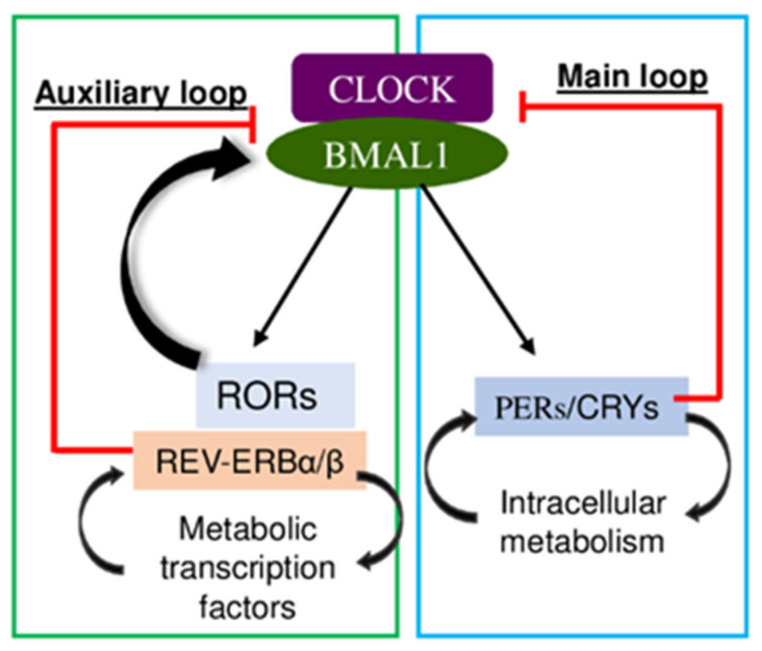
*Clock* and *Bmal1* drive expression of two negative-feedback arms controlled by Period/Cryptochrome (*Per*/*Cry*) and the two paralogs, *Rev-Erbα/β* and *ROR*. In turn, these negative-feedback arms repress Bmal1/Clock transactivation function (Per/Cry) or Bmal1 expression (Rev-Erbα/β) and Bmal1 activation (ROR).

**Figure 4 ijms-23-12954-f004:**
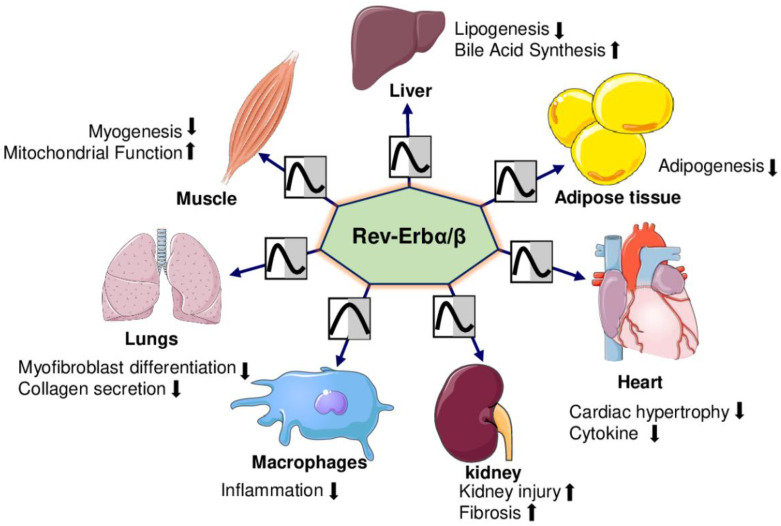
Rev-Erbα/βs display robust 24 h oscillations and regulate the physiological functions of the organs. Rev-Erbs reduce adipogenesis and lipogenesis and increase bile acid synthesis in adipose tissue and liver, respectively. Rev-Erbs reduce cardiac hypertrophy and inflammatory cytokines in the heart, myofibroblast differentiation, and collagen production in the lungs. Rev-Erbs decrease myogenesis and increase mitochondrial function in skeletal muscles. Rev-Erbs increase injury and fibrosis in the kidney. (
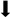
 = decrease, 
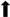
 = increase).

**Figure 5 ijms-23-12954-f005:**
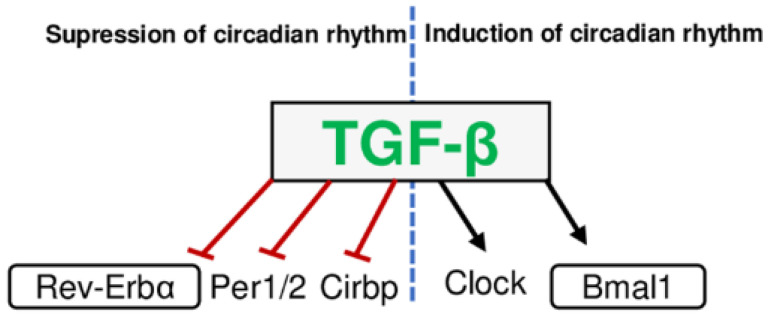
TGF-β inhibits the expression of the negative regulators of circadian clocks (*Rev-Erbα/β*, *Per1/2*, *Cirbp*) and prolongs the wake period. In addition, TGF-β reverses the wake period by stimulation of *Clock* and *Bmal1* expression.

**Table 1 ijms-23-12954-t001:** Role of Rev-Erb and their synthetic ligands in tissue fibrosis.

**A**				
**Effector Organ**	**Rev-Erb**		**Functions**	**Reference**
Lung	Knockout mice		Increased pulmonary myofibroblast activating markers such as collagen-1 and αSMA Increased neutrophil lung influx and proinflammatory cytokines (IL-6, MCP-1)	[99,100]
	Lower mRNA and protein expression		Smokers and COPD patients	[101]
	Reduced *Rev-Erbα* mRNA and protein expression		Emphysema mice	[102]
Liver	Knockout mice		Plasma VLDL concentrations and APOC-III expression Hepatic steatosis and reduced bile acid synthesis	[55,103,104]
	Knockout of *Rev-Erbα/β* mice fed with HFD		Hepatic triglyceride levels	[105]
	Knockout in primary hepatocytes		* CYP4A * expression, lipid accumulation, and oxidative stress	[106]
Adipose tissue	Knockout mice		Increased plasma adiponectin and its expression in WAT	[107,108]
			Adiposity and hypertrophied adipocytes	[107,108]
Skeletal muscle	*Rev-Erbα/β* double-knockout mice		Altered lipid metabolism and circadian wheel-running behavior	[109]
	Knockout mice		Reduced running capacity Reduced skeletal muscle mass and muscle fiber cross-section area Reduction of the regenerative capacity of muscle	[59,110,111]
	Upregulation in Mice		Loss of muscle mass	[110]
	Upregulation in C2C12 myocytes		Increases exercise capacity, mitochondrial content, and activity	[59]
	Upregulation in myoblasts		∙ Reduces dexamethasone-induced atrophy-related genes	[110]
Kidney	*Rev-Erbα/β* knockout mice		Decreased sensitivity to folic acid-induced acute kidney injury and diminished rhythm in disease severity	[112]
	*Rev-Erbβ* knockout mice		Increasing gene and protein expressions of *Slc7a11* and *HO1*	[112]
**B**				
**Rev-Erb ligands**	**Effector** **organ**	**Animal Model**	**Effects**	**Reference**
SR9009 (Agonist)	Heart	TAC mice	Prevented cardiomyocyte hypertrophy, reduced fibrosis	[113]
		Mice	Improved left ventricles (LV) function and survival after myocardial infarction Decreased expression of cytokines IL-6, MCP-1, MMP9, and immune cells (neutrophil and proinflammatory macrophages) infiltration into the infarcted heart Reduced ratio of proinflammatory M1 macrophages to anti-inflammatory M2 macrophages	[97]
		LDL-receptor deficient mice fed with a western diet	Reduced atherosclerosis	[98]
GSK4112 (Agonist)	Lungs	Human small airway epithelial cells and mouse lung fibroblasts	Attenuated both LPS and cigarette smoking-induced inflammatory response and reduced pulmonary fibrosis	[100,114]
		HFL-1 cells	Suppressed TGF-β–induced fibroblast differentiation	[115]
SR9009 (Agonist)	Liver	Mice	Induced weight loss and reduced plasma triglycerides and cholesterol	[63]
			Inhibition of cholesterol biosynthesis	[116]
		HSC	Inhibition of HSCs proliferation	[117]
		Mice	Prevented alcohol-induced liver injury	[106]
		NASH mice	Reduced hepatic inflammation (IL-1α, IL-1β, Ifnγ, and TNFα) and fibrosis	[118,119]
		Ccl4-induced fibrosis in mice and Rat HSCs	Reduced fibrosis	[120,121]
SR6452 (Agonist)	Adipose tissue	3T3L1 cells	Induction of adipocyte differentiation	[122]
SR9011 (Agonist)		Mice	Reduced fat mass and lipogenic gene expression	[58]
SR9009 (Agonist)		Mice	Induced weight loss and reduced plasma lipids	[58]
SR6452 (Agonist)	Skeletal muscle	Mice	Regulation of atrophy-related genes	[110]
SR8278 (Antagonist)		Dystrophic mice	Improved mitochondrial biogenesis and muscle functions	[123]
GSK1362 (Inverse agonist	Kidney	HEK293 cells	Inhibition of chemokines and cytokines	[114]
SR8278 (Antagonist)		Mice	Reduced kidney damage	[112]

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
