# Peer review of "Role of Circadian Transcription Factor Rev-Erb in Metabolism and Tissue Fibrosis"

_ijms, 2022, doi:10.3390/ijms232112954_

Round 1

Reviewer 1 Report

Review of manuscript ijms-1928177

Role of circadian transcription factor Rev-erb in metabolism and tissue fibrosis

Authors

Ghulam Shere Raza, Nalini Sodum , Yagmur Kaya , Karl-Heinz Herzig

The manuscript of Raza et al is the well written review devoted to a problem of regulatory role of Rev-Erb circadian transcriptor factor on metabolism as well as fibrosis of different organs. The main text is preceded by introduction characterizing the mechanism or fibrosis and circadian rhythm. Both TGF-beta and HIF-alfa were described in details and their effects on circadian clock genes were discussed. In later chapters the role  of Rev-Erb effect on fibrosis and metabolism of different organs (heart, liver, lungs, adipose tissue and skeletal muscle) were explained. The final part of the manuscript contains correctly written conclusion

The manuscript could be interesting for the readers of International Journal of Molecular Sciences.

There are a few concerns with the study:

Minor concerns

  1. In introduction the characteristics of myofibroblasts should be completed with full information about myofibroblasts markers.
  2. It is not clear why for more detail discussion only TGF-beta and HIF-alfa were selected from many mediators participating in fibrotic process.
  3. In table 1B in first column:It should be added what role of each compound plays: antagonistic or agonistic. The present form is confusing.
  4. The Authors should consider replacement of the title of the final chapter from Conclusions into Summary and conclusions.

Author Response

We thank the referees for their thoughtful comments to improve the quality of our manuscript.

The response to the reviewer comments are attached as word file.

Reviewer 2 Report

The paper by Raza et al. provides an overview on the impact of circadian transcription factor Rev-Erb changes in metabolism and tissue fibrosis.  The topic is not new and has been the object of numerous prior reports, and I am not sure it would find many readers.

However, while the first part of the paper, related to general control of circadian rhythm and influence of TGF and HIF-1a is a very standard University  thesis, the following section, related specifically to tissue fibrosis, in particular considering new agents affecting Rev-Erb, is definitely of higher interest and more original.  Unfortunately the Authors tend to go back and forth, first describing the general aspects of, eg liver fibrosis, first examined as mechanims of disease, then from the point of view of a Rev-Erb agonist,  then again as cellular aspects of Rev-Erb.

This kind of approach is confusing to the reader and does not provide the reader with a conclusion he can advantageously grasp.

I would suggest the following:

1)Cut the initial part to 1/10, just summarizing knowledge that is widely available

2) follow a well standardized scheme for organ disease:  role of Rev-Erb;  agonism or antagonism with specific agents (The Table is quite close to what is needed;  consequences of agonism or antagonism in different conditions)

3) since this is supposed to be a medical journal, please give specific space to potential clinical indications of these agents.  In your confusing approach you mention rat data, then clinical data and than back to in vitro and so on.  The case is most strking in the adipose tissue section.

In conclusion, the paper reflects some effort by the authors.  These however did not do much in order to make it readable and informative.

Author Response

We thank the referees for their thoughtful comments to improve the quality of our manuscript.

The reviewer response is attached as word file

Round 2

Reviewer 2 Report

The Authors of this review article have done work on it, giving it a more logical sequence and providing  a more critical overview.

I am, however, still irritated by the poor handling of the English language.  The Authors seem not to know  the difference between singular and plural.

A few examples:

Line 62 : myofibroblast synthesize

121:  circadian clock regulate

231:  left ventricles

270:  hepatocytes number

and many others.

Otherwise the paper tried to provide what could be new to a reader with some exceptions.  I see no reason to give chapters 3 and 4, ie TGF-beta an HIF-1alpha.  These are very well known topics and just a few lines would suffice.

Coming to the text, the only original part, in my view, is the list of drug molecules, that are given, somewhat cursorily, in the text.  Thus their presentation as a table is OK.  Instead I see no reaon to provide in Table format the different tissues, well reported in the text. Specifically, in the text, I see  a somewhat confusing presentation of adipose tissue (“debilitated adipocytes?), there is probably little reason to specify features of collagen metabolism before the discussion on the role of Rev-Erb.

I believe the Authors can do some more work on the text before it reaches adequate standing

Author Response

Dear Reviewer, 

Kindly find attched response to the comments.
